# Rare and declining bird species benefit most from designating protected areas for conservation in the UK

A. E. Barnes [1], J. G. Davies [2], B. Martay[2], P. H. Boersch-Supan [1], S. J. Harris[1], D. G. Noble[1], J. W. Pearce-Higgins [1,3] & R. A. Robinson [1] ✉

There have been recent renewed commitments to increase the extent of protected areas to combat the growing biodiversity crisis but the underpinning evidence for their effectiveness is mixed and causal connections are rarely evaluated. We used data gathered by three large-scale citizen science programmes in the UK to provide the most comprehensive assessment to date of whether national (Sites of Special Scientific Interest) and European (Special Protection Areas/Special Areas of Conservation) designated areas are associated with improved state (occurrence, abundance), change (rates of colonization, persistence and trend in abundance), community structure and, uniquely, demography (productivity) on a national avifauna, while controlling for differences in land cover, elevation and climate. We found positive associations with state that suggest these areas are well targeted and that the greatest benefit accrued to the most conservation-dependent species since positive associations with change were largely restricted to rare and declining species and habitat specialists. We suggest that increased productivity provides a plausible demographic mechanism for positive effects of designation.

The current high rate of biodiversity loss is one of the biggest global environmental issues, interacting with others to exceed environmental planetary boundaries[1,2]. One approach to address this is to protect an increasing area of land and sea from anthropogenic threats[3,4]. Globally, the world has barely met the United Nations Convention of Biological Diversity Aichi target 11 of at least 17% of terrestrial and inland water being designated for protection by 2020 (ref. [5]). Furthermore, there is a high degree of variation between countries[6] and effective implementation of the targets has been challenging overall[7]. Despite these shortcomings, the draft new Post-2020 Biodiversity Global Framework includes an increased ambition to "ensure that at least 30 per cent globally of land areas and of sea areas are conserved through effectively and equitably managed, ecologically representative and well-connected systems of protected areas and other effective area-based conservation measures"[8]. While protected areas (PAs) vary in their aims, which include goals related to ecosystem services and contributions to people's livelihoods, there are, at root, three factors that determine their effectiveness for improving the status of biodiversity: (1) coverage, that is, how much and what biodiversity is included within PAs and how representative it is; (2) improved population status of focal species or habitat condition, that is, are PAs being managed well and external pressures minimized; and, more generally, (3) can a network of PAs collectively support the restoration or expansion of wider populations/habitats of conservation concern, even outside of their boundaries?

Given this diversity of outcomes and wide variation in what protection means on the ground in terms of associated management

[1]British Trust for Ornithology, The Nunnery, Thetford, UK. [2]British Trust for Ornithology (Scotland), Unit 15 Beta Centre, Stirling University Innovation Park, Stirling, UK. [3]Conservation Science Group, Department of Zoology, Cambridge University, Cambridge, UK. ✉e-mail: rob.robinson@bto.org

practices, metrics to measure effectiveness can be difficult to construct[9] and evidence for the effectiveness of PAs is mixed[10,11]. PAs often target areas of greater species diversity and concentrations of species of conservation concern[12] but not always successfully[13,14]. Regarding their impact, measures of PA extent can sometimes be positively associated with biodiversity trends, as measured by species diversity[13] and population abundance trends[15–17], although not always[18–21]. Furthermore, as species distribution changes lag behind those of climate[22], PAs increasingly have a role in allowing populations to adapt to changing climates[23,24], although with high variability between species[25]. Whether associations between PAs and biological responses are a function of protection per se or underlying patterns of land use and habitat type associated with their selection is often unclear[16]; crucially, the variation in species responses to PAs is largely unexplained. The causal links between PA and conservation outcomes are rarely tested[26]. We used a comprehensive assessment of the impacts of statutorily designated PAs on the larger part of the UK avifauna to address this lack of understanding around the underlying processes to better maximize the delivery of PAs for biodiversity conservation.

Designating PAs is a relatively straightforward policy tool to address biodiversity losses; implementing these effectively is, of course, a different matter[27]. The large-scale, citizen science-based biodiversity monitoring undertaken in much of Europe[28] provides an opportunity to quantify these wider benefits of designated area networks. Birds are among the best-studied taxa, with many species of high conservation concern and therefore the target of protection individually. In the UK, 29% of species are regarded as being of high conservation concern[29], with protection offered primarily by sites designated under either national (Sites of Special Scientific Interest (SSSI)) or European (Natura, 2000) legislation. SSSIs are given some protection against damaging operations and planned developments, whereas for Natura sites Member States are only obligated to take appropriate steps to avoid the effects of pollution or deterioration subject to an economic interest test; both largely fall into the International Union for Conservation of Nature (IUCN) Protected Area Management Category IV[30]. As is increasingly common[31], these designations overlap and while SSSIs aim to protect representative habitats in a geographical area, Special Protection Areas (SPAs), under Directive EC/14/2009, and Special Areas of Conservation (SACs), under Directive EC/43/1992, are targeted at the 'best' locations for, respectively, particular bird species and biodiversity/habitats more generally. The status of these PAs is measured through the 'Common Standards Monitoring' (CSM) protocol[32], which uses a standard typology to assess the condition of the features for which each PA is designated (factor 2 above), but which cannot, feasibly, capture the wider benefits (factor 3).

We used data gathered from three large-scale, citizen science programmes to test whether, across most of the UK avifauna, PAs are associated with (1) higher probabilities of occurrence and greater abundances, that is, a better biodiversity 'state' and whether they are associated with positive changes in that, that is, (2) greater probabilities of persistence (equivalently, lower extinction risk) or colonization and/or (3) more positive (or less negative) trends in abundance. Further, we might expect (4) that PAs targeted at (particular) bird species (SPAs) have a greater positive effect on bird populations generally than those designated for other biodiversity/environmental features (SACs). Importantly, in doing these comparisons, we controlled for differences in land cover, elevation and climate to increase the likelihood of responses being directly a function of PA status. We also tested (5) whether the variation in response between species is linked to changes in breeding success, a key potential mechanism. Given that we were assessing an entire avifauna, we expected a mix of responses, so we then identified the species that the extent of a PA most benefits, specifically testing (6) whether PAs benefit species that are rare, have declining population trends or are habitat specialists (often those of most conservation concern[33]). Finally, (7) we considered whether the

communities in areas with greater PA extent are more diverse, more specialist or provide a refuge for cold-adapted species, testing their relevance for climate change adaptation. We took two approaches to control for the effect of confounding environmental variation: a regression-based approach and statistical matching[34]. Because the bird monitoring data were collected at a 1 km (abundance, productivity) or 2 km (occupancy) scale (Methods), we estimated the effect of the proportion of a PA in each sample square for each species, using generalized additive mixed models, predicting more positive results with greater PA presence in the sample square, while including covariates of land cover, geographical location, elevation, climate and human population density. Second, we compared a matched sample of squares with no or some (>10%) PA coverage weighted by their similarity on the basis of these environmental variables. Each approach has advantages and disadvantages and answers slightly different questions, so we regarded the commonality of inference between the two approaches as an important test of the validity of our results.

## Results

### Species occur more frequently and in greater abundance

Many species occurred more frequently, and more abundantly, in areas with a greater extent of PA (Fig. 1); the matched analysis yielded similar results but with generally more positive responses (Extended Data Fig. 1). While there were a wide range of individual species responses, 48% of species had a significant positive association (compared to 21% negatively) between their likelihood of occurrence and the extent of the PA (number of species with significantly positive responses versus the number species with significantly negative species: $\chi^2 = 17.1$; $P < 0.001$; Supplementary Table 1), with a positive mean association between occurrence and PA extent (mean slope = 0.49 ± 0.07; Supplementary Table 2). The species with strong negative responses to PAs tended to be those that are common in urban areas (Supplementary File 1). Similarly, the abundances of 48% of species were significantly positively associated with PA extent ($\chi^2 = 8.6$, $P = 0.003$), again with an overall positive mean association (0.25 ± 0.05). Thus, there was support for our first hypothesis, that is, species occur more often and more abundantly where there is a greater extent of PA.

### Changes in occurrence but not abundance are positive

Although the absolute number of species showing significant positive (25%) and negative (26%) associations between colonization and PA extent was similar ($\chi^2 = 0.04$, $P = 0.83$; Fig. 1), there was a significant positive overall response, with species more likely to colonize tetrads with a greater extent of PA (mean effect = 0.27 ± 0.08; Supplementary Table 2), reflecting particularly strong positive effects for a number of rare/localized species (see below). Species were also significantly more likely to persist in sites with a greater extent of PA (0.23 ± 0.09; Supplementary Table 2), with a tendency for more species to have significantly positive (30%) than negative (20%) effects (Supplementary Table 1). These effects were largely repeated in the matched analysis (Extended Data Fig. 1).

We found no evidence for a significant effect of PA on abundance trends (Fig. 1), with a similar number of species (20% versus 23%) having significant positive and negative effects (Supplementary Table 1) and an overall mean effect that did not differ from zero (Supplementary Table 2). Thus, there was evidence that range dynamics (the balance of colonization and local extinction; hypothesis 2) but not changes in abundance (hypothesis 3), were more positive in PAs.

### Effectiveness varies with the reason for designation

As predicted, given our focus on bird species, these patterns of association were strongest with SPA designation, with a greater number of species being more likely to occur (41% significantly positive versus 30% significantly negative) or have higher abundances

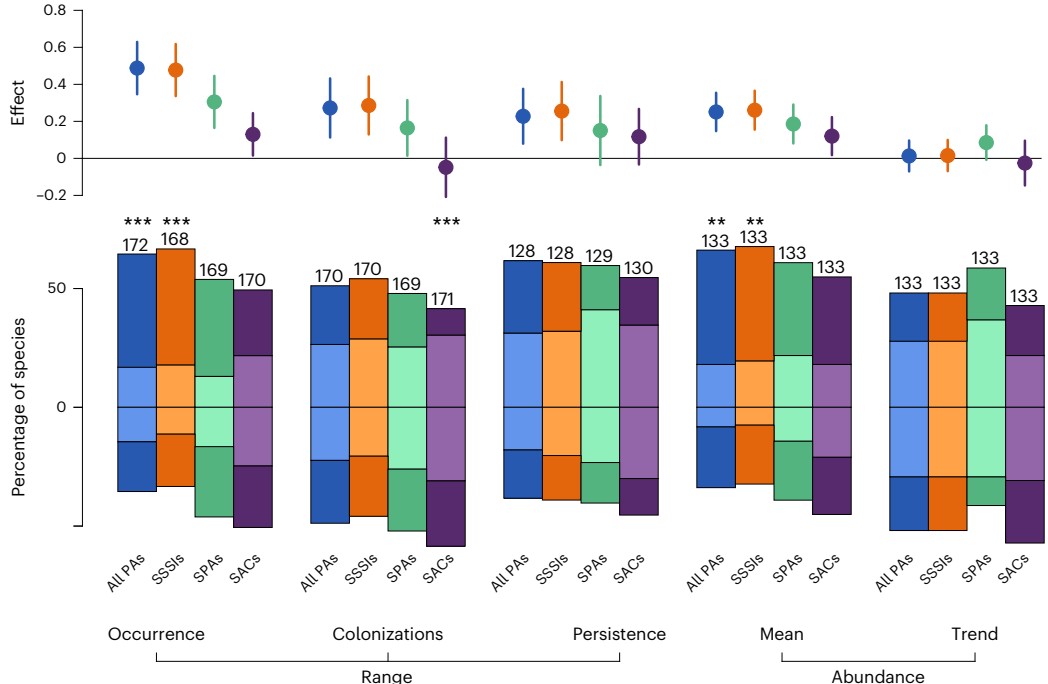

**Fig. 1 | PA designation influences range and abundance dynamics of individual bird species.** The bars (bottom) represent the percentage of species and the points (top) represent the mean (and 95% confidence intervals (CIs)) of the effect sizes among individual species with negative and positive associations between the population measure (occurrence, colonization, persistence, abundance and trend in abundance) and percentage cover of PA within the monitored square. In the bar chart, species with a significant relationship with the different designations are shown in dark colours while species with a non-significant relationship are shown in light colours. The numbers indicate the sample size for each. The asterisks indicate whether the two-sided test of proportion showed a significantly different number of species with significant positive effects compared to negative effects. **$P < 0.01$, ***$P < 0.001$. See Supplementary Table 1 for the $P$ values.

(39% versus 25%), with increasing SPA extent (Fig. 1). In contrast, similar numbers of species were more or less likely to occur with increasing SAC extent (Supplementary Table 1). Furthermore, the mean relationships for occurrence, colonization and abundance trend were significantly stronger in SPAs than SACs (Supplementary Table 2). These results are thus consistent with our fourth hypothesis, that the most effective PAs for birds were those designated specifically for birds.

### Positive effects are linked to higher productivity

Overall, variation in reproductive success between Constant Effort Sites (CES) sites, for the subset of species with productivity data, was negatively correlated with PA extent although this effect was least marked in relation to SPA extent (Supplementary Tables 1 and 2). On sites containing SPAs (but not SSSIs or SACs), those species that exhibited higher productivity with greater PA extent were also those that had higher abundances with more PA (Fig. 2a and Supplementary Table 3). Furthermore, those species for which productivity tended to increase more over time in PAs also tended to show more positive abundance trends with greater PA extent (Fig. 2b and Supplementary Table 3). Thus, comparison of two independent datasets provides support for our fifth hypothesis that higher productivity is associated with more positive trends in abundance at least for SPAs.

### Rare and habitat specialist species benefit most

After accounting for body mass and phylogenetic relatedness, and weighting estimates to reduce the influence of species with uncertain responses, positive relationships between the extent of PA and occurrence, colonizations, persistence and abundance were most apparent for rarer species (those with lower population size) and habitat specialists (Fig. 3). Furthermore, species that were declining nationally had more positive (or less negative) trends in abundance in sites with greater PA extent (Fig. 4), a relationship that was stronger with SPA than SAC extent (Supplementary Table 4). Similarly, occurrence and persistence of species that were legally protected or of conservation concern were higher and significantly more so than for unlisted/green species where there was greater PA extent (Supplementary Table 5). However, the effect of PA extent on abundance of listed species was less marked and when their generally smaller population size was accounted for, there were fewer significant differences between the species groupings, although the overall pattern of benefit remained (Supplementary Table 6). The matched analysis gave similar results (Extended Data Fig. 2) albeit with generally smaller effect sizes. Hypothesis (6) was therefore supported with habitat specialists and rare (and declining) species most positively associated with PAs.

Wetland and woodland species were both more likely to occur and persist in sites with a greater extent of PA and occur in higher abundances, while species associated with urban environments were less likely to do so (Fig. 3 and Supplementary Table 4). Wetland, but not woodland, species also showed more positive abundance trends with greater PA extent, while urban species occurred at lower abundances but also with more positive population trends.

### Bird communities are more specialist and cold-adapted

Overall, species richness was generally lower where there was more PA but sites with greater PA coverage supported more specialist and more cold-dwelling species (Fig. 5). They also experienced reductions in species diversity over time and a shift towards more cold-dwelling communities. Thus hypothesis 7 is partially supported in that communities in areas with greater PA cover are more specialist and cold-adapted but they are not more diverse.

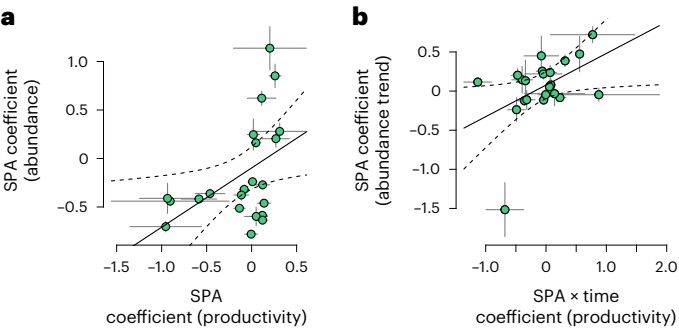

**Fig. 2 | Relationship between productivity (CES) and abundance (BBS) PA coefficients. a**, Productivity model SPA coefficients against abundance model SPA coefficients. **b**, Productivity model SPA × time coefficients against abundance model SPA × time coefficients, with s.e. bars, in both cases *n* = 22 species (details in Supplementary Table 3). The outlier in **b** is Cetti's warbler; excluding this point results in the significance becoming marginal (*β* = 0.19 ± 0.11, *P* = 0.099).

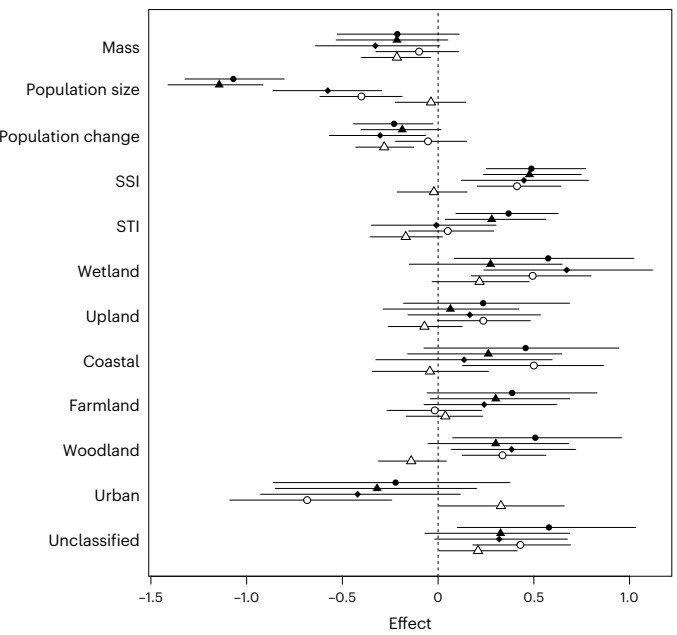

**Fig. 3 | The effect of PA designation on range and abundance dynamics varies with ecological traits.** The extent to which the relationship between range (closed) and abundance (open) population measures and PA extent varies depending on species traits, from a phylogenetically weighted regression. The parameter estimates (±95% confidence limits) are, respectively, occurrence (closed circles, *n* = 172 species), colonization (closed triangles, *n* = 170), persistence (closed diamonds, *n* = 128), mean abundance (open circles, *n* = 133) and abundance trend (open triangles, *n* = 133). Mass, population size and population change are log-transformed values of mass, population size and population change. Species Specialisation and Temperature Indices are the Species Specialisation and Temperature Indices. The final seven traits refer to the habitat in which species are most commonly found (Supplementary File 1).

## Discussion

Through our comprehensive assessment, we highlight a range of associations that are consistent with the PA network having had a positive impact on bird conservation over the last three decades in one of the least biodiverse nations with significant shortfalls in PA effectiveness[11]. We found strong evidence that PAs were originally well targeted for birds, in that bird species occurrence and abundance is higher on PAs, particularly on SPAs. Evidence that PA positively influence bird outcomes is more mixed: colonizations and persistence are higher in PAs but abundance trends are not. Specifically, for rarer, declining or habitat specialist species, PAs were associated with higher probabilities of occurrence and lower rates of extinction. Furthermore, specialists were more abundant and declining species had less negative trends in abundance, strongly suggesting that the benefits of this network are greatest for species most in need of conservation action. In the context of previously uncertain biodiversity responses to PAs and global ambitions to increase PA extent to address the current biodiversity crisis, these headlines strengthen the expectation that by achieving the ambitious target of 30% terrestrial and freshwater protected area coverage countries can make an important contribution to addressing the global biodiversity crisis[35] but also emphasize the importance of appropriately targeting and managing them.

By controlling for large-scale variation in land cover, topography, human population pressure and climate, we showed that species were not only more likely to occur in PAs, over and above the surrounding land characteristics (a much debated question[18]), but further show that PAs are also effective in positively altering species dynamics, particularly of those species of most conservation interest (that is, those with smaller or declining population sizes). It is difficult to completely separate the effects of protection and land cover because, by definition, PAs target particular habitats; for instance, wetland species were almost universally associated with PAs since wetlands are a particularly threatened and hence protected habitat in the UK. Statistical matching can alleviate, although not eliminate, this problem; this complementary approach yielded similar conclusions. We provide evidence for an underlying mechanism of these positive effects on demography. Thus, those species with the most positive effects of PA on their status and trend also showed higher rates of breeding success in PAs. There is growing evidence in support of management interventions being effective in boosting the breeding success of birds of conservation concern[36,37], contributing to positive associations between those species and protected areas[25,36,38], and the potential to stem or reverse species declines more generally[39]. The lack of a positive relationship between

productivity and PA extent and abundance trend across species (Supplementary Table 3) suggests either that PAs are not associated with greater habitat quality (and many are in 'unfavourable' condition[11]) or, given that they tend to be associated with greater rates of occurrence and higher abundance, there may be density-dependent limits to productivity in PAs.

The effects were strongest for SPAs, that is, areas specifically designated under European legislation for protecting birds, particularly rarer and declining habitat specialists. This supports the results of continent-wide associations[40] and previous single-species analyses[36,38]. Thus, the positive effects of PA extent were most apparent for species associated with woodland and wetland habitats, both relatively rare and fragmented natural or semi-natural habitats in the British countryside[41] that have been the target of much conservation effort. Importantly, the effects we found were present despite wide variation in the intensity of site management of the PA[11], which we did not account for. We may have found more pronounced effects had we been able to account for the differing habitat quality of these sites, which is likely to be as important as their size and quantity. Assessing the impact of PA management across a network of sites is challenging, as the difficulties around implementing CSM attest; however, broad-scale citizen science is unlikely to provide the necessary resolution to robustly test these and is better suited to quantifying broader impacts, as we did in this study. The pattern of increased abundance of urban species is indicative of wider increases in generalist species[42] and outside pressures on PAs generally. While the lack of a general relationship with abundance trend may indicate that PAs are not being appropriately managed, the interpretation of such patterns is complex and requires detailed consideration[43].

We also show that responses at the species level scale up to alter bird communities, with PAs associated with reduced diversity and more negative diversity and evenness trends, potentially driven by complex responses across species since not all threatened habitats in the UK support high species richness or diversity (for example, see Sullivan et al.[42]). Associations between PA extent and metrics of habitat (community specialization index (CSI) and community temperature index (CTI)) specialization show that areas with a greater PA extent support communities that tend to consist of more habitat specialists and cold-adapted species. Furthermore, rates of increase in CTI, a key signal of climate change impacts on bird communities[44], are reduced in areas with a greater extent of PA, suggesting that PAs have played a role in ameliorating these impacts. Similarly, analyses of breeding bird data from Finland show that declines in retreating northern species were lower in PAs than outside[45] and that in the UK, local extinctions of northern bird species at low elevations/latitudes were reduced by PAs[25]. Interactions between temperature-related community changes and either PA status[46] or the extent of semi-natural habitat[47,48] provide further evidence that PA networks can modify community-level responses to climate change, particularly by facilitating climate-driven colonization of new sites[25,49,50].

We provide an unusually comprehensive assessment to date of the effects of protected sites on a national avifauna. We document significant positive responses, particularly for rare species and habitat specialists of conservation concern, which are impacting bird communities in those PAs and potentially increasing their resilience to impacts of climate change; we also demonstrate the potential for existing large-scale structured biological surveillance data to monitor and evaluate their broader effectiveness. While we have also provided unique evidence linking the potential benefit of PAs to greater relative breeding success, further work is required to assess the extent to which the simple protection of rare habitats is sufficient. In the context of habitats that are otherwise being lost outside PAs, this alone could account for a positive effect but many of the species considered in this study (such as those that are rare and/or declining) are also subject to active management, especially on PAs, further contributing to the positive responses. At a time of debate about the need to expand the coverage of global PA from the current level of around 17% to 30% by 2030, these findings provide strong evidence to support the contention that such a policy would be likely to deliver significant biodiversity benefit and contribute to species recovery as part of the IUCN Green Status for many species and not necessarily only those for which sites are designated[51]. The fact that responses were greatest for the SPA network (that is, targeted at protecting bird habitats) suggests that to maximize the effectiveness of any new PA networks, new networks need to be targeted towards the species and habitats that are most threatened.

## Methods

### Data sources

**Species occurrence, colonization and persistence.** We estimated species breeding occurrence, colonization and persistence from two nationwide Atlas surveys of the UK avifauna undertaken in 1988–1991 (ref. [52]) and 2007–2011 (ref. [53]). Volunteer surveyors recorded the presence of each species in each of 42,561 and 46,390 2 × 2 km squares (tetrads) in the two Atlas periods; 29,851 of these tetrads (of a possible 61,843) were surveyed in both periods[54]. The tetrads covered the whole of the UK and are subdivisions of a national 10 km grid (with 25 tetrads per 10 km square); there was at least some coverage within each 10 km square, except in Northern Ireland where tetrads were surveyed from within every second 10 km square. Coverage was generally higher in areas with higher human population density (Supplementary Fig. 1).

Species occurrence, colonizations and persistence were assigned using presence and absence from the Atlas data. Species were classified

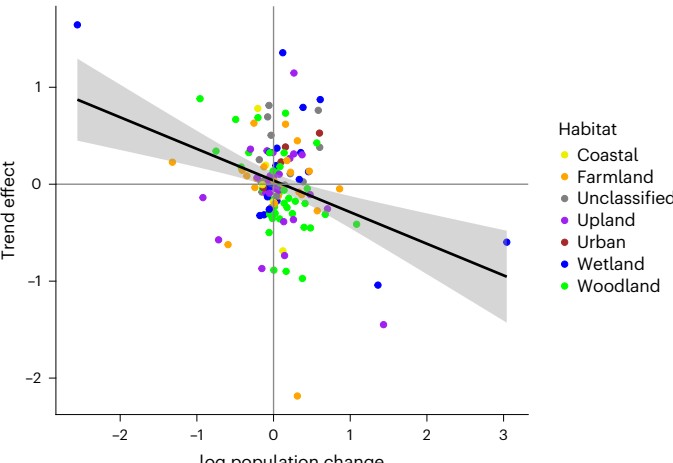

**Fig. 4 | Declining species have more positive population trends where there is greater PA extent.** Each point is a species estimate coloured by habitat preference (n = 133), including the linear regression line (Supplementary Table 4) and 95% confidence interval (shaded grey).

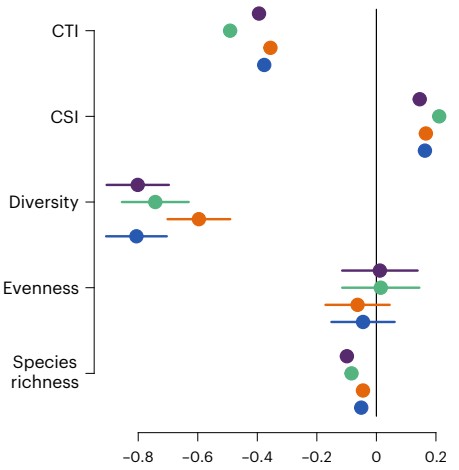
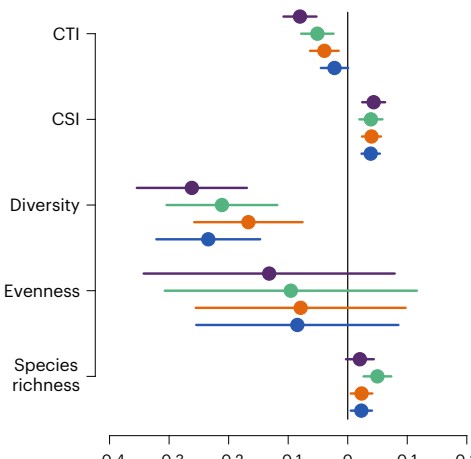

**Fig. 5 | Community structure is influenced by PA designation.** The relationship between the extent of PA and metrics of community structure (species richness, evenness, diversity, Community Specialisation Index and Community Temperature Index (left) and trends in these (right; n = 5977). The relationship with all protected areas, SSSIs, SPAs and SACs, are shown in blue, orange, green and purple, respectively; the error bars represent the 95% confidence limits of the parameter estimates.

as occurring in a tetrad if it was recorded in either survey period. Species were classified as colonizing if they were absent in a square in the 1988–1991 Atlas but present in the 2007–2011 Atlas; thus, only squares for which no presence of the species was recorded in the early Atlas were included in this analysis. Species were classified as persistent if they were present in both the 1988–1991 and the 2007–2011 Atlas, so only squares with the species in question present in the early Atlas were included in this analysis. Persistence is the complement of extinction rate (that is, persistence = 1 − extinction) which we used to ensure that positive estimates had a consistent interpretation across metrics.

Species sightings were designated as possible, probable or confirmed breeders. To exclude birds that may not have been breeding birds, we excluded sightings from any tetrad that had no probable or confirmed breeders of that species within their containing 10 km square; 241 species met this criterion but we excluded non-native species and species which occurred in fewer than 20 tetrads, leaving 180 species (Supplementary File 1).

**Species abundance and trend.** Species abundance (and population trend) data were derived from the annual British Trust for Ornithology (BTO)/Joint Nature Conservation Committee (JNCC)/Royal Society for the Protection of Birds (RSPB) Breeding Bird Survey (BBS) for the period 1994–2019 (ref. [55]). Briefly, volunteer surveyors recorded all adult birds they saw or heard on two 1 km line transects traversing a 1 km square on two visits during the breeding season (early visit: 1 April–15 May; late visit: 16 May–30 June). Squares were selected according to a stratified random design that accounted for the number of volunteers available in each of 83 geographical regions, with a total of 6,718 squares covered (Supplementary Fig. 1, increasing from 1,570 squares in 1994 to 4,005 surveyed in 2019). Our measure of square-level annual abundance was the maximum count of each species from the two visits to each square in a year. We considered 133 species (Supplementary File 1) recorded in an average of at least 100 squares per year over the period (1994–2019); as above, we excluded non-native species and records of likely non-breeding species (for example, the fieldfare *Turdus pilaris*, flocks of waders). Seabirds, except gulls and terns, were also excluded due to poor coverage of their coastal breeding habitat in BBS squares. A small number of sites in upland areas (approximately 100) included an adjacent square (so a 2 km transect) to maximize the number of records in poorly covered areas with a low overall density of birds, which we accounted for by including the number of squares (1 or 2) as an offset in the models to standardize for coverage effort.

**Productivity and productivity trend.** We estimated productivity (number of young birds fledged per adult) from a constant effort mark–recapture programme (CES[56]) for the years 1990 (when 97 sites operated) through to 2019 (114 sites), with a total of 490 sites (Supplementary Fig. 1). Briefly, volunteers erected mist nets in set positions for a set length of time on, usually, 12 visits through the breeding season. The total number of juveniles caught relative to the number of adults in each year provides an index of overall productivity for the site and immediately surrounding area. Capture totals for a site were omitted from the dataset if fewer than four early (from the first six) and four late (from the last six) visits were made at a site in any given year, to minimize the effect of any missing visits[57], or if fewer than ten juveniles and adults of a species were caught in a year. A total of 22 species were included (Supplementary File 1).

**Designated areas and environmental data.** The location and extent of designated areas (Extended Data Fig. 3) were obtained from the Natural England Open Data Geoportal (https://naturalengland-defra.opendata.arcgis.com/), the Scottish spatial data portal (https://spatialdata.gov.scot/geonetwork/srv/eng/catalog.search), the Welsh Lle Geo-portal (https://lle.gov.wales/catalogue) and Open Data NI (https://www.opendatani.gov.uk/); all were accessed on 1 November 2020. We extracted

shapefiles for SSSIs, SPAs and SACs and calculated the proportion coverage within the land area of each 1 km square (abundance, trend, productivity) or 2 km square (occurrence, colonizations, persistence). Obtaining definitive designation dates (many of which will pre-date our dataset since about 50% of the UK network had been designated by 1974 (ref. [18])) is difficult due to alterations in site boundaries over time and the lag between designation and management starting. Thus, we treat all sites as designated for the duration of our time period. Each square therefore has a variable amount of PA coverage and we predicted that those squares with a larger amount of PA would show more a positive effect, which we estimated using a generalized additive mixed model (GAMM), which is described below.

We extracted habitat data from the Land Cover Map 2015 (1 km percentage aggregate class from Great Britain and Northern Ireland)[58,59]. The aggregate land cover classes (and percentage cover) are: broad-leaved woodland (7.4); coniferous woodland (4.9); arable (24.7); improved grassland (32.7); semi-natural grassland (8.0); mountain, heath and bog (10.4); saltwater (0.7); freshwater (1.2); coastal (2.1); and built-up areas and gardens (8.0). Mean elevation was calculated from the ASTER Global Digital Elevation Model v.003 (ref. [60]) for each 1 km cell. Human population density (in 2015) was obtained from the Global Human Settlement Layer dataset[61] and we calculated mean population density (ind km$^{-2}$) within a 10 km radius as a mesoscale measure of human influence.

**Species traits.** Body mass is broadly correlated with many aspects of life history and was used as a proxy for these (Supplementary File 1). Mean body mass for all species was taken from Robinson[62]. Legal protection is afforded to species on Schedule 1 of the Wildlife and Countryside Act (1981, as amended) at a national scale and on Annex 1 of the Directive on the Conservation of Wild Birds (EC/14/2009, the Birds Directive) at a European scale. Conservation status was taken from the first Birds of Conservation Concern (BoCC) list[63], which categorized species into three categories according to their, then, perceived vulnerability in relation to population size, range and abundance trend as: green (least concern), amber and red (highest concern). Population size in the early 1990s and late 2010s was derived from the work of the Avian Population Estimates Panel[64,65] and the national population change was taken as the ratio of these two numbers. The primary habitat each species occurred in was taken from Gibbons et al.[52] and the degree of habitat specialization of each species using the species specialization index (SSI) of Sullivan et al.[42].

**Data analysis**
**Overall approach.** First, for each species and population metric (for example, occupancy, abundance), we fitted a GAMM (described below) to estimate the relationship between the population metric and the area of designated land within a 1 or 2 km survey square, while accounting for variation in habitat and climate. The coefficients from these individual species models for occurrence (and changes in this through colonizations and persistence) and abundance (and linear trend in this over time) were then analysed using four general linear models (GLMs) for each population metric. The first three GLMs each had a single response variable of each type of conservation status traits (BoCC, Annex 1, Schedule 1) since we were interested in the importance of PAs for these designated species. We then fitted a fourth GLM to explore the role of underlying ecological traits in determining the strength of a species response to the extent of the designated area. In this last model, the species-specific effect estimates were weighted by the inverse of their variance to give greater weighting to those species that were estimated with more confidence. All analyses were carried out in R v.4.0 (Ref. [66]).

In all these analyses we initially investigated how population metrics varied in relation to the area of designated land (of any type) within a survey square and then repeated the analyses three times, using the area of SSSI, SPA and SAC as the response variables.

The location of PAs is non-random, with more PAs in upland areas further away from human habitation[67]. We accounted for this by including relevant confounding covariates in our analysis but an alternative approach is to create a statistical counterfactual by 'matching' treatment and non-treatment sites on the basis of similarity in these (or other) covariates[68–70]. Therefore, we also undertook an analysis where we 'matched' (for each PA type separately) squares with PA (defined as those with >10% coverage) with similar squares with no PA coverage. Matching was based on the same covariates used in the regression analysis, using Mahalanobis distance matching without replacement and without any callipers in the matchit() function of the MatchIt package v4.3.4[71]. We used partial, rather than full, matching to enable calculation of robust s.e. to appropriately propagate the uncertainty through to the traits analysis (see below). Matching for all PA types reduced the imbalance in coverage for Atlas (occupancy) and BBS (abundance), generally to within standardized mean differences <0.25, but not CES (productivity) datasets (Supplementary Fig. 3); it also reduced sample sizes for the BBS as a result of their being insufficient 'control' squares (Extended Data Fig. 1). Twenty-eight per cent of all PA squares had a matched control with 14–27% for the individual designations. Between 36 and 60 species were removed from the BBS matching analysis. Because imbalances remained for some covariates, we used covariate adjustments in the subsequent analysis of the matched sample. This analysis was performed in parallel to the main analysis (for the Atlas and BBS parts) using the same steps but with the matched rather than full dataset; however, note that it is testing a slightly different hypothesis using a binary variable of PA presence rather than a continuous one of extent).

**Species models.** Measures of bird occurrence, colonization, persistence, abundance, abundance trend and productivity for each bird species (where appropriate; see below) in each square and in each year were modelled using GAMMs in the mgcv package v1.8-40[72]. We accounted for variation in climate by including a tensor smooth function of elevation, easting and northing; weather by including year (as a factor) as a random effect (in the abundance and productivity models); and habitat by including a linear functions of nine habitat types. (We excluded the arable category to avoid overfitting and parameter identifiability issues since the habitat coverage would otherwise sum to 1.) We also included human population density and its square since counts peaked at intermediate densities. For the abundance models we also included a quadratic function of year (continuous) to account for any overall long-term changes in the population size. Our focus was then on the linear term for the proportion of each survey square that was designated and, for the abundance and productivity analyses, the interaction of this term with (linear) year as a measure of the influence of PA extent on trends in these over time.

Species occurrence, colonizations and persistence were all binary variables that we modelled with a binomial distribution and a logit link function. For some species, sample sizes for the persistence models were very small because squares were only included in the analysis if they were monitored in both Atlas periods and there was a presence of the species in the first Atlas period. Therefore, we could not model persistence for 50 species because they had smaller sample sizes than the number of coefficients in the model ($n < 140$). Likewise, for eight species colonization models failed to run because of very few colonization events. Models were assessed using the gam.check() function of mgcv and the functions provided in the DHARMa package v0.4.4[73]; species with overdispersed and zero-inflated models were excluded. We also excluded species for which the parameter estimates were extreme outliers compared to parameter estimates for other species because this was likely to indicate poorly fitting models. In general, species with extreme parameter estimates were also the species with overdispersion so were already excluded. The number of species for which the models were successful varied depending on the type of PA

considered as an explanatory variable; however, occurrence, colonization and persistence models could be run for 168–172 species (of the 180 for which we had data; see above), 169–171 species (of the 172 for which we had sufficient colonization events) and 129–130 species (of the 130 species for which we had sufficient persistence data), respectively depending on PA type.

Abundance (and trend therein) was modelled with a negative binomial distribution (and a log link function) since Poisson models generally exhibited substantial overdispersion. Model fit was assessed using the gam.check() function in the mgcv package[72]. We fitted models for all 133 species.

For productivity, the proportion of a year's CES captures that were juvenile was modelled as a binomial process with a logit link in an events–trials formulation, where each juvenile individual counted as a 'success'[56]. We fitted these models for 22 species.

**Summarizing the responses.** We summarized the correlation between species population measures (that is, occurrence, colonization, persistence, abundance and trend) and the proportion of the square which was designated in two ways. First, for each of the population measures, we compared the number of species with significantly positive associations with an area of designation (and each type of designation separately) to the number of species with significantly negative associations using a one-sample binomial test. Second, for each of the population measures, we compared the mean across species of the associations with the area of designation (and each type of designation separately) using t-tests. We then compared the response of species to SPAs and SACs by using paired t-tests to compare the association between species population measures and SPA area with the same association with SAC area.

**Traits analysis.** To determine which traits were associated with a stronger positive response to PA extent, we fitted linear models with the extent of a designated area coefficient (from the previous analysis for the individual species models) as the response variable and measures of conservation concern or ecological traits as the explanatory variables. To account for phylogenetic relatedness between species, we used an Ericson phylogenetic tree averaged from 1,000 trees downloaded from https://birdtree.org/ (Jetz et al.[74], accessed 8 March 2021) and performed a phylogenetically weighted regression using the MCMCglmm v2.32[75] and ape v5.5[76] packages. We used inverse Wishart priors for the covariance structure of the residuals ($V = 1$, $nu = 0.002$) and the random effects ($V = 1$, $nu = 1$), respectively. For all other parameters, the MCMCglmm default of an improper flat prior was used. A single chain was run with 50,000 iterations with a burn-in sample of 5,000 iterations that were discarded and every 25th subsequent sample retained to produce an adequate sample. We visually checked model diagnostics using trace and density plots of fixed and random effect estimates.

We fitted four models for each type of designation, each with different covariates: the three measures of conservation concern were analysed as three separate models and included log population size in a separate run to account for the fact that more common species tended to have more precise estimates and hence weighted more heavily in the analyses. A fourth model contained all the ecological traits (log body mass, log population size and change, SSI, species temperature index (STI) and habitat indicator status/association). We did not do a traits analysis on productivity because there were too few species.

**Community analysis.** We treated community metrics similarly to the species measures (described above) in the sense that we had one measure per BBS square per year derived from the species recorded in a given square and year. Before constructing the community indices, we corrected the abundance measure by a species detectability

factor[77] to provide a more comparable measure of relative abundance across species. We considered three measures of community structure: species richness (number of species recorded); diversity (Hill's $N_2$ (refs. [78,79])); and evenness (diversity divided by richness). We also considered two synthetic trait measures, the CSI[80] and CTI[81]. The CSI is the density-weighted mean of the individual SSI for species occurring in a given square and measures the tendency for wildlife communities to increasingly consist of generalist species. SSI was calculated for each bird species as the coefficient of variation of the density of a species across 12 dominant habitat classes across all BBS squares[42]. Similarly, the CTI is the density-weighted average of individual STIs, the long-term average temperature over the species range, for which we used values derived from the full European breeding range[44]. For each of these metrics, we fitted GAMs with appropriate distributions and land cover, climate variables along with the extent of the designated area and its interaction with (linear) year.

## Reporting summary

Further information on research design is available in the Nature Portfolio Reporting Summary linked to this article.

## Data availability

The datasets generated and/or analysed during the current study are available in the Supplementary Information (in the *figshare* repository (https://doi.org/10.6084/m9.figshare.20200895). The raw data are available upon request from the BTO.

## Code availability

All code was analysed in R using open-source packages and functions and can be accessed via GitHub at https://github.com/BritishTrustForOrnithology/BirdsOnProtectedAreas.

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

## Acknowledgements
The BTO/JNCC/RSPB Breeding Bird Survey is a partnership jointly funded by the BTO, RSPB and JNCC. The Constant Effort Scheme was jointly funded by the BTO and JNCC, with fieldwork conducted by volunteers. The Atlas projects were funded through a combination of corporate and governmental sponsorship and charitable donations from members and supporters of the non-governmental organizations that coordinated the projects (BTO, BirdWatch Ireland and the Scottish Ornithologists' Club). We thank all the volunteers for their efforts over many years. This work was funded jointly by the JNCC, NatureScot, Natural England, Nature Resources Wales and the Department of Agriculture, Environment and Rural Affairs, Northern Ireland through the Terrestrial Surveillance Development and Analysis partnership with BTO. We thank D. Allen, B. Eardley, N. Newton, A. Nisbet and R. Weyl for their support and comments on earlier drafts, along with those of G. Buchanan, D. O'Brien, H. Hoskins, F. Sanderson and two anonymous referees.

## Author contributions
J.W.P-H., D.G.N. and R.A.R. conceived the study. A.E.B. undertook the analyses of BBS data and wrote the first draft, B.M. analysed the Atlas data, J.G.D. analysed the CES data and P.H.B-S. designed the matched analysis. S.J.H. organized the BBS and managed those data. All authors contributed critically to the final draft.

## Competing interests
The authors declare no competing interests.

## Additional information
**Extended data** is available for this paper at https://doi.org/10.1038/s41559-022-01927-4.

**Correspondence and requests for materials** should be addressed to R. A. Robinson.

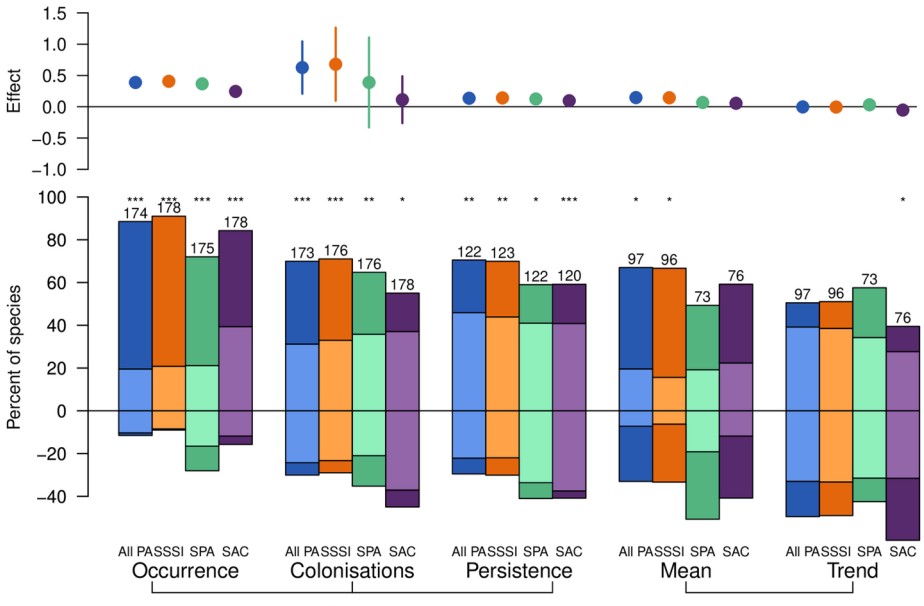

**Extended Data Fig. 1 | Overall species results of the matched analysis.** The bars (bottom) represent the percent of species and the points (top) represent the mean (and 95% confidence intervals, those for the abundance metrics are too small to show) of effect sizes among individual species with negative and positive associations between the population measure (occurrence, colonisation, persistence, abundance and trend in abundance) and percentage cover of protected area within the monitored square. In the barchart, species with a significant relationship with the different designations are shown in dark colours while species with a non-significant relationship are shown in light colours. Numbers indicate the sample size for each. Asterisks whether the two-sided test of proportion is significantly different number of species with significant positive effects compared to negative effects: * p < 0.05; ** p < 0.01; *** p < 0.001. See Supplementary Tables 7, 8 for P-values.

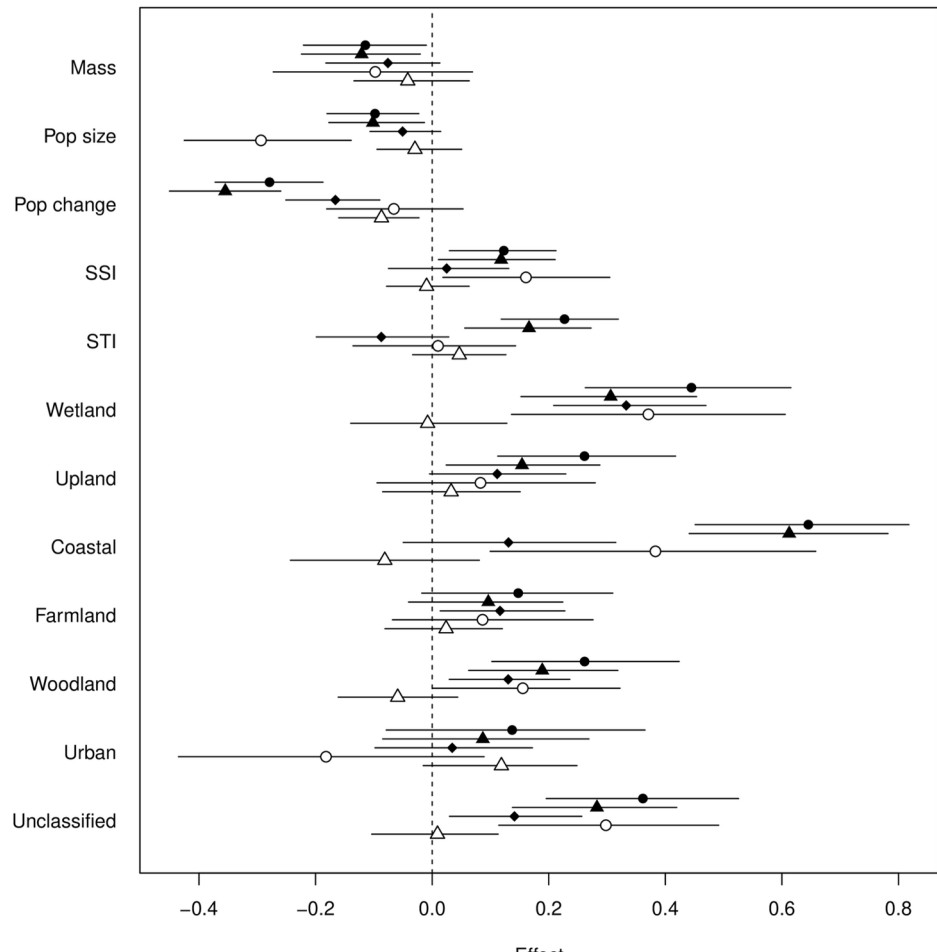

**Extended Data Fig. 2 | How the effect of PAs on range and abundance dynamics using a matched analysis varies with ecological traits.** The extent to which the relationship between range (closed) and abundance (open) population measures and PA extent varies depending on species traits, from a phylogenetically-weighted regression carried out using an MCMCglmm. The mean estimates (± 95% confidence limits) are, respectively, occurrence (closed circles, n = 174), colonisation (closed triangles, n = 173), persistence (closed diamonds, n = 122), mean abundance (open circles, n = 133) and abundance trend (open triangles, n = 133). Mass, population size and population change are log-transformed values of mass, population size and population change. SSI and STI are the Species Specialisation and Temperature Indices. The final seven traits refer to the habitat in which species are most commonly found (Suppl File 1).

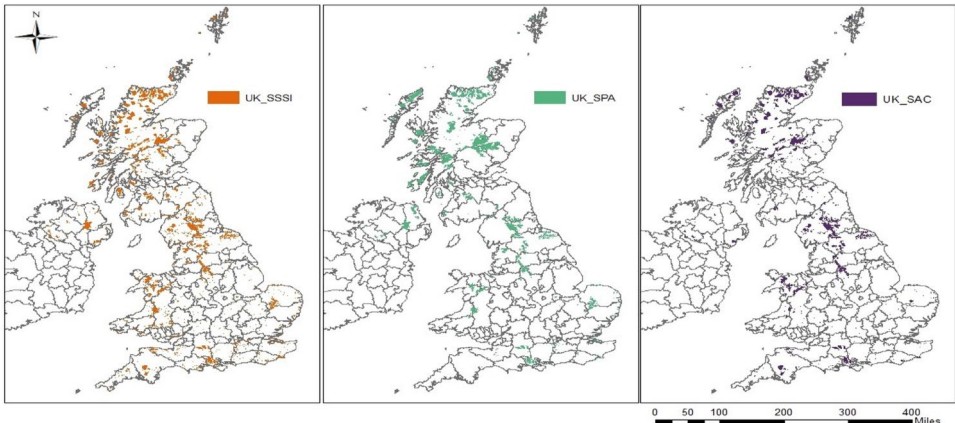

**Extended Data Fig. 3 | Conservation designation coverage.** Maps of the three types of designated area in the UK. Contains Ordnance Survey data © Crown Copyright and database right 2022.

# Reporting Summary

## Statistics

For all statistical analyses, confirm that the following items are present in the figure legend, table legend, main text, or Methods section.

| n/a | Confirmed | |
|---|---|---|
| ☐ | ☒ | The exact sample size (*n*) for each experimental group/condition, given as a discrete number and unit of measurement |
| ☐ | ☒ | A statement on whether measurements were taken from distinct samples or whether the same sample was measured repeatedly |
| ☐ | ☒ | The statistical test(s) used AND whether they are one- or two-sided *Only common tests should be described solely by name; describe more complex techniques in the Methods section.* |
| ☐ | ☒ | A description of all covariates tested |
| ☐ | ☒ | A description of any assumptions or corrections, such as tests of normality and adjustment for multiple comparisons |
| ☐ | ☒ | A full description of the statistical parameters including central tendency (e.g. means) or other basic estimates (e.g. regression coefficient) AND variation (e.g. standard deviation) or associated estimates of uncertainty (e.g. confidence intervals) |
| ☐ | ☒ | For null hypothesis testing, the test statistic (e.g. *F*, *t*, *r*) with confidence intervals, effect sizes, degrees of freedom and *P* value noted *Give P values as exact values whenever suitable.* |
| ☐ | ☒ | For Bayesian analysis, information on the choice of priors and Markov chain Monte Carlo settings |
| ☐ | ☒ | For hierarchical and complex designs, identification of the appropriate level for tests and full reporting of outcomes |
| ☐ | ☒ | Estimates of effect sizes (e.g. Cohen's *d*, Pearson's *r*), indicating how they were calculated |

*Our web collection on statistics for biologists contains articles on many of the points above.*

## Software and code

Policy information about availability of computer code

| | |
|---|---|
| Data collection | No software was used in data collection, only data storage (Oracle) as the data were collected by citizen science volunteers. |
| Data analysis | All data was analysed in R using open sourced packages and functions and can be accessed on request via GitHub, https://github.com/BritishTrustForOrnithology/BirdsOnProtectedAreas. |

For manuscripts utilizing custom algorithms or software that are central to the research but not yet described in published literature, software must be made available to editors and reviewers. We strongly encourage code deposition in a community repository (e.g. GitHub). See the Nature Portfolio guidelines for submitting code & software for further information.

## Data

Policy information about availability of data

All manuscripts must include a data availability statement. This statement should provide the following information, where applicable:
- Accession codes, unique identifiers, or web links for publicly available datasets
- A description of any restrictions on data availability
- For clinical datasets or third party data, please ensure that the statement adheres to our policy

The datasets generated during and/or analysed during the current study are available in the Suppl File 1 (in figshare) repository, https://doi.org/10.6084/m9.figshare.20200895. The raw data is available on request from BTO.

# Human research participants

Policy information about <u>studies involving human research participants and Sex and Gender in Research.</u>

| | |
|---|---|
| Reporting on sex and gender | This data was not collected |
| Population characteristics | N/A |
| Recruitment | N/A |
| Ethics oversight | N/A |

Note that full information on the approval of the study protocol must also be provided in the manuscript.

# Field-specific reporting

Please select the one below that is the best fit for your research. If you are not sure, read the appropriate sections before making your selection.

☐ Life sciences ☐ Behavioural & social sciences ☒ Ecological, evolutionary & environmental sciences

For a reference copy of the document with all sections, see nature.com/documents/nr-reporting-summary-flat.pdf

# Ecological, evolutionary & environmental sciences study design

All studies must disclose on these points even when the disclosure is negative.

| | |
|---|---|
| Study description | Using volunteer collected survey data (see below) to assess whether measures of bird species were influenced by the proportion of protected area and what type of species (using traits) were mainly affected. |
| Research sample | Bird Atlas surveys of the UK avifauna undertaken in 1988-91 (Gibbons et al. 1993) and 2007-11 (Balmer et al. 2013), a total of 61,843 2-km squares.<br>Annual BTO/JNCC/RSPB Breeding Bird Survey data for the period 1994-2019 (Freeman et al. 2007) for a total of 6718 sites.<br>Constant effort mark-recapture program of birds (CES, Robinson et al. 2009) for the years 1990 (when 97 sites operated) through to 2019 (114 sites), with a total of 490 sites. |
| Sampling strategy | Sample sizes were chosen based on the relevant datasets, e.g. colonization from the Atlas dataset had to have the birds present in the second Atlas but not the first, and the removal of non-native species and seabirds (see below), and whether the species models converged during analysis. We used two approaches for analysis: i) a modelling approach that used all relevant data and included covariates to account for potentially confounding variables between PA and ii) statistical matching that used the same confounding variables to match PA sites with non-PA sites most similar to them. During the matching process sample sizes were reduced for the BBS and CES dataset as a result of their being insufficient 'control' squares to match to the "treatment" squares that contained PA. |
| Data collection | Atlas: volunteer surveyors record all adult birds seen using the 2-km square, in two timed visits (minimum 1 hour) in the breeding season (early visit - April - May, late visit - June - July).<br>BBS: volunteer surveyors record all adult birds they see or hear on two, 1km line-transects traversing a 1km square on two visits in the breeding season (early visit - 1st April-15th May and late visit - 16th May-30th June)<br>CES: volunteers erect mist-nets in set positions for a set length of time on, usually, 12 visits through the breeding season to capture all birds (adults and juveniles) at the site. |
| Timing and spatial scale | As above. |
| Data exclusions | Non-native bird species and seabirds were removed from the analysis as they do not report on them for trends and we cannot accurately census seabirds from these surveys. |
| Reproducibility | N/A: not an experiment |
| Randomization | N/A: not an experiment |
| Blinding | N/A: not an experiment |

Did the study involve field work? ☐ Yes ☒ No

# Reporting for specific materials, systems and methods

We require information from authors about some types of materials, experimental systems and methods used in many studies. Here, indicate whether each material, system or method listed is relevant to your study. If you are not sure if a list item applies to your research, read the appropriate section before selecting a response.

## Materials & experimental systems

| n/a | Involved in the study |
|-----|----------------------|
| ☒ | Antibodies |
| ☒ | Eukaryotic cell lines |
| ☒ | Palaeontology and archaeology |
| ☐ | ☒ Animals and other organisms |
| ☒ | Clinical data |
| ☒ | Dual use research of concern |

## Methods

| n/a | Involved in the study |
|-----|----------------------|
| ☒ | ChIP-seq |
| ☒ | Flow cytometry |
| ☒ | MRI-based neuroimaging |

## Animals and other research organisms

Policy information about studies involving animals; ARRIVE guidelines recommended for reporting animal research, and Sex and Gender in Research

| | |
|---|---|
| Laboratory animals | Study did not involve laboratory animals. |
| Wild animals | The presence of wild birds were observed and recorded or captured and ringed in field sites across the UK by skilled volunteers following https://www.bto.org/our-science/bto-approach-science/animal-research-ethics; all animal capturing was undertaken by ringers by licensed by BTO, on behalf of the relevant statutory country body |
| Reporting on sex | This information was not collected for observations (BBS and Atlas) but was collected in the CES data, but not used in this analysis as productivity was per adult (male or female). |
| Field-collected samples | The study did not involve samples collected from the field. |
| Ethics oversight | All applicable national guidelines for the care and use of animals were followed according to https://www.bto.org/our-science/bto-approach-science/good-scientific-practice |

Note that full information on the approval of the study protocol must also be provided in the manuscript.

