## [Peer Review File · Nature Ecology & Evolution]

Peer Review Information

Journal: Nature Ecology & Evolution

Manuscript Title: Rare and declining bird species benefit most from designating protected areas for conservation in the UK

Corresponding author name(s): R.A. Robinson

Editorial Notes:

Reviewer Comments & Decisions:

Decision Letter, initial version:
--

28th March 2022

Dear Dr Robinson,

Your Article, "Do conservation designations provide positive benefits for bird species and communities?" has now been seen by 2 reviewers. You will see from their comments copied below that while they find your work of considerable potential interest, they have raised quite substantial concerns that must be addressed. In light of these comments, we cannot accept the manuscript for publication, but would be very interested in considering a revised version that addresses the serious concerns of Reviewer 2 around statistical matching.

We hope you will find the reviewers' comments useful as you decide how to proceed. If you wish to submit a substantially revised manuscript, please bear in mind that we will be reluctant to approach the reviewers again in the absence of a major revision of the manuscript that includes a re-analysis with matching.

I would also like to note that Reviewer 1 commented to editors that most of the statistical methods were outside their area of expertise, so we are particularly interested in seeing your responses to Reviewer 2's comments.

I have attached Reviewer 1's marked up copy of your manuscript as a doc file.

If you choose to revise your manuscript taking into account all reviewer and editor comments, please highlight all changes in the manuscript text file.

* Include a "Response to reviewers" document detailing, point-by-point, how you addressed each referee comment. If no action was taken to address a point, you must provide a compelling argument. This response will be sent back to the referees along with the revised manuscript.

* If you have not done so already we suggest that you begin to revise your manuscript so that it conforms to our Article format instructions at <http://www.nature.com/natecolevol/info/final-submission>. Refer also to any guidelines provided in this letter.

[REDACTED]

If you wish to submit a suitably revised manuscript we would hope to receive it within 6 months. If you cannot send it within this time, please let us know. We will be happy to consider your revision so long as nothing similar has been accepted for publication at Nature Ecology & Evolution or published elsewhere.

Nature Ecology & Evolution is committed to improving transparency in authorship. As part of our efforts in this direction, we are now requesting that all authors identified as 'corresponding author' on published papers create and link their Open Researcher and Contributor Identifier (ORCID) with their account on the Manuscript Tracking System (MTS), prior to acceptance. This applies to primary research papers only. ORCID helps the scientific community achieve unambiguous attribution of all scholarly contributions. You can create and link your ORCID from the home page of the MTS by clicking on 'Modify my Springer Nature account'. For more information please visit www.springernature.com/orcid.

Please do not hesitate to contact me if you have any questions or would like to discuss the required

2revisions further.

Thank you for the opportunity to review your work.

[REDACTED]

Reviewer expertise:

Reviewer #1: spatial conservation prioritisation, UK conservation

Reviewer #2: protected area effectiveness

Reviewer comments:

Reviewer #1 (Remarks to the Author):

This is a good manuscript with an important message supported by robust analyses. The message that protected areas work and that targeted protected areas work better is of interest to conservation practitioners and policy makers.

The manuscript sets out to answer several research questions using robust citizen science datasets curated by the authors, which would not be possible for most other regions of the world. The datasets employed therefore provide a unique opportunity to study the relationship between protected area designation and avifauna occurrence, abundance, persistence, colonisation, population trends and breeding success. The authors also investigate whether protected areas benefit species that are rare and declining or habitat specialists over more diverse species and whether they provide a refuge for cold-adapted species in the face of climate change.

The manuscript provides some strong scientific evidence for the effectiveness of protected areas using robust datasets and statistical methods. The manuscript is quite heavy on the statistics, which is not surprising given the strong expertise of the authors in this field, but it would benefit from a little more policy context to make the findings easier to interpret for a wider audience. It would be good to mention Common Standards Monitoring as the mechanism by which Country Nature Conservation Bodies (CNCBs) in the UK monitor protected areas. CNCBs have been struggling to keep up with the CSM reporting cycle (around half of SSSIs have had no formal assessment in the past six years) – does citizen science data such as the Bird Atlas and BBS provide a partial solution?

The study assesses all species for which there are sufficient data, but SPAs and SACs are designated for specific interest features/habitats and it is these that are reported on under the Directives. SSSIs can be designated for biological or geological interest features; does it make sense to treat geological SSSIs in the same way as biological SSSIs in this analysis? You analyse the proportion of survey squares (also referred to as “sites”) covered by protected areas. I wondered whether it matters

3therefore that the total protected area extent is not factored in.

Overall, this manuscript has well-defined research questions and uses appropriate datasets and statistical methods to answer them. It would be helpful to restructure the results section so that it more clearly follows the research questions i-vii set out in the introduction. Of particular relevance is the finding that targeting protected areas matters; clearly, this is the case for SPAs and birds, and this has implications for the taxonomic representativeness in the designation of additional protected areas under future targets.

I have included a commented version of the manuscript with some more specific comments e.g. missing references, a few suggested word changes etc.

Reviewer #2 (Remarks to the Author):

This is an exceptionally well-written and clear study exploring the association between protected area (PA) designation and the state, change, community structure, and demography of birds in the UK. The topic is of relevance in light of the currently ongoing negotiations for a post-2020 biodiversity framework, where some are pushing for expanding the PA coverage target to 30 % by 2030. Clearly, we urgently need more evidence of the role of PAs and how effective PAs are as a policy tool to protect habitats and avoid population declines and extinctions. Most such studies have used forest loss or land cover change as a proxy for PA effectiveness, evaluating the impact PAs have in avoiding land use change. More studies are needed that look at the effect on species, communities, and population trends. The submitted study therefore sounded extremely interesting based on the abstract. The authors use Atlas based data on birds in the UK to develop a wide set of indicators for population viability: exploring both state, change, community structure, and even demography. This is interesting. However, unfortunately the study design in terms of evaluating PA effectiveness is not sufficient and is merely exploring associations, rather than attempting to infer cause-effect. Powerful methods to create quasi-experimental setups are available and applying matching to the dataset could result in some interesting results. The need to account for the non-random location of PAs become apparent when looking at Figure S2, showing that PAs in the UK (as in most other countries in the world) are situated "high and far" (see seminal paper by Joppa & Pfaff 2009 calling for the need to account for this in PA effectiveness studies). There have been recent advances applying matching to population data (Terraube et al. 2020 (cited); Jellesmark et al. 2022 (preprint)), so should be doable with the data in the study as the atlas data is available from both inside and outside PAs. The likely most important confounding factors to account for would be accessibility and remoteness from urban centers (in addition to all commonly explored factors). Unfortunately, I think we have moved beyond the point of associative studies for inferring whether a policy intervention works or not, we need more robust study designs that can explicitly answer how well conservation interventions work (Ferraro & Hanauer. 2014; Baylis et al. 2016; Schleicher 2020).

Minor comments:

line 8-9: The sentence "but the underpinning evidence for their effectiveness is mixed with causal connections rarely evaluated" gives the reader the impression the study will apply a counterfactual

4study design, i.e. evaluating what would have happened, had the PAs not been established.

line 33-34: The phrasing of the third objective remains unclear for me: "iii) can they collectively facilitate the restoration or expansion of wider populations/habitats of conservation concern?"

line 52: please specify early on which national avifauna? Now I think one needs to go all the way to the Methods to see it written out that it is UK.

line 55: perhaps include the current first draft of the new targets as a reference.

<https://www.cbd.int/doc/c/abb5/591f/2e46096d3f0330b08ce87a45/wg2020-03-03-en.pdf>

line 91: as said before the paper is exceptionally clear and well written, but here as a first time reader, I suggest the authors open up what they mean with "in areas with ", as that otherwise is only further explained in the methods and it would be good if the main text could stand on its own.

References

Baylis et al. 2016. Mainstreaming impact evaluation in nature conservation. *Conservation Letters* 9: 58– 64.

Ferraro & Hanauer. 2014. Advances in measuring the environmental and social impacts of environmental programs. *Annual Review of Environment and Resources* 39: 495– 517.

Joppa & Pfaff 2009. High and Far: Biases in the Location of Protected Areas
<https://doi.org/10.1371/journal.pone.0008273>

Jellesmark et al. 2022 (preprint)
<https://www.biorxiv.org/content/10.1101/2022.01.14.476374v1.abstract>

Schleicher et al. 2020. Statistical matching for conservation science
<https://conbio.onlinelibrary.wiley.com/doi/full/10.1111/cobi.13448>

Terraube et al. 2020. Assessing the effectiveness of a national protected area network for carnivore conservation <https://www.nature.com/articles/s41467-020-16792-7>

Author Rebuttal to Initial comments

We thank the reviewers for their constructive comments, we have replied to each point in **bold** below and have highlighted these changes in the track-changed version of the document. All references to line numbers are to the track changed version.

Reviewer #1 (Remarks to the Author):

This is a good manuscript with an important message supported by robust analyses. The message that protected areas work and that targeted protected areas work better is of interest to conservation practitioners and policy makers.

We thank the referee for recognising the value of our work and our presentation of it

The manuscript sets out to answer several research questions using robust citizen science datasets curated by the authors, which would not be possible for most other regions of the world. The datasets employed therefore provide a unique opportunity to study the relationship between protected area designation and avifauna occurrence, abundance, persistence, colonisation, population trends and breeding success. The authors also investigate whether protected areas benefit species that are rare and declining or habitat specialists over more diverse species and whether they provide a refuge for cold-adapted species in the face of climate change.

The manuscript provides some strong scientific evidence for the effectiveness of protected areas using robust datasets and statistical methods. The manuscript is quite heavy on the statistics, which is not surprising given the strong expertise of the authors in this field, but it would benefit from a little more policy context to make the findings easier to interpret for a wider audience. It would be good to mention Common Standards Monitoring as the mechanism by which Country Nature Conservation Bodies (CNCBs) in the UK monitor protected areas. CNCBs have been struggling to keep up with the CSM reporting cycle (around half of SSSIs have had no formal assessment in the past six years) – does citizen science data such as the Bird Atlas and BBS provide a partial solution?

We appreciate the referee's recognition of the strength of our work and have tried provide the context requested by including reference to CSM in the introduction (L79) and discussion (L238) but also improved reference to global policy (such as the CBD 30x30 (L31) goals and IUCN Green Status) as requested below.

The study assesses all species for which there are sufficient data, but SPAs and SACs are designated for specific interest features/habitats and it is these that are reported on under the Directives. SSSIs can be designated for biological or geological interest features; does it make sense to treat geological SSSIs in the same way as biological SSSIs in this analysis? You analyse the proportion of survey squares (also referred to as "sites") covered by protected areas. I wondered whether it matters therefore that the total protected area extent is not factored in.

As the referee points out, each site is designated with a specific feature in mind. Often this is a particular (set of) bird species but often not (either other biological species, habitats or geological features), however protection to maintain other features may affect bird populations and our aim is to assess the impact of the policy tool, i.e. network as a

6whole, so we prefer to include all sites, particularly as a common element is protection from damaging anthropogenic activities. We thus recognise that there may be benefits beyond the narrow set of species for which individual sites are designated, contributing to the improved status of species and communities more generally. Because we may expect some differences between sites designated for birds compared to other features (not just geological) we explicitly analyse SPA (birds) and SAC (other species and habitats) separately, indeed this is one of our key hypotheses (Hypothesis iv, L141).

We don't analyse "the proportion of survey squares covered by PA" rather the proportion of *each* survey square that is PA. We have clarified this in the methods (L352) and, since it is critical to understanding our results, we have added a statement to the introduction to emphasise this point (L98).

Overall, this manuscript has well-defined research questions and uses appropriate datasets and statistical methods to answer them. It would be helpful to restructure the results section so that it more clearly follows the research questions i-vii set out in the introduction. Of particular relevance is the finding that targeting protected areas matters; clearly, this is the case for SPAs and birds, and this has implications for the taxonomic representativeness in the designation of additional protected areas under future targets.

The structure of the results section was designed to mirror the research questions set out in the introduction, but with the questions phrased in a more concise way to facilitate reading - we have added direct reference to the individual hypotheses to each subhead to clarify this.

I have included a commented version of the manuscript with some more specific comments e.g. missing references, a few suggested word changes etc.

L10: These tend to be designated for interest features/species; is that worth mentioning in the text?

This is a good point, and we mention this at L76 in the main text.

L11: Does this compare to BirdLife's state, pressure, response model for monitoring IBAs? Thinking of potential links to the global framework.

We don't think so, we are simply grouping the types of response we look at, these largely fall within the 'state' category of BirdLife's classification.

L16: It might be appropriate to include a reference somewhere to the IUCN Green Status of Species (Akçakaya et al., 2018).

Thanks for pointing this out, we have included a more recent reference (Grace et al. 2021) to this concept in the discussion (L281).

L50: Not in references

Reference corrected (L59)

L60: Not in references

Reference added (L604)

L88: It would be helpful to aid comprehension if the results were structured so that they follow research questions i-vii above.

We have included reference to each hypothesis, see above

L93: It would be good to see a brief explanation for why 20% of spp. have negative association with PAs.

We have added some text (L119) these are mostly species not associated with PA habitats, particularly those found in urban areas.

L94: I understand this to be the extent of PA within the survey square, rather than the full extent of the PA itself. Maybe good to make that more explicit, what implications does that have for the study?

That is correct, we have added clarifications of this important point at L98 and 352. This relates to the use or otherwise of statistical matching, discussed below.

L109: Suggest putting this under a separate subheading so that it follows i-vii above.

We prefer to keep this as, since change in status (hypothesis i) may occur through either a change in frequency, or in abundance, as discussed here

L112: I suggest that what is meant here is “extirpation”

We have clarified our meaning as local extinction (L138)

L126: What is a “site” in this context? Survey squares?

A site in this context is a CES site, we have clarified this (L152)

L136: Is this explained somewhere?

Our approach to determining the impact of traits on PA response (i.e. which species benefit) is described at L454-67

L138: What is the definition of “rare” species here? Is there a threshold for “low” population size?

We are describing the direction of the regression rather than a threshold and have made this clearer by rewording to use comparative terms (L165)

L140: Meaning survey squares and PA extent within those?

This is correct

L166: Technically the UK has high terrestrial PA coverage of around 29%. Suggest replacing “coverage” with “effectiveness” here.

Done (L194).

L192: It would be very interesting to look at PA condition, but not easy to align with survey cells.

It would indeed be very interesting, but this is a much bigger and more involved analysis than we can present here.

L219: I suggest that what is meant here is “extirpation”

Local extinction is the commonly used term we believe.

L264: I think the reason you refer to 10km squares and tetrads in one sentence is that one refers to the Atlas and the other to the BBS, but could you make this more explicit for the reader? Otherwise it could be confusing.

Both refer to Atlas sampling, we have added some text to clarify this at L294, 309

L282: Brief explanation for how this was offset in the models?

We have added some text to explain the offset, L328

L300: Bird counts may be within a different part of the square to the PA, although at this scale perhaps that is OK.

Yes, they may be, however birds are quite mobile so it is likely that even some coverage of PA may have a beneficial effect since birds in a square may use it for part of the time. Indeed this gives us greater power to assess the effects of PA, since one might expect greater effects where PA covers a greater proportion of the square. This underpins the regression we took and we have added some text to clarify this (L352)

L306: If there are fractional classes data then they might reveal relationships not discernible from the aggregate classes, but a lot more work; not suggesting you need to do that here.

We agree with this comment, but our primary focus was on the effect of PA not modelling detailed habitat relationships per se and agree that it is not appropriate in this context.

L310: Topographic roughness might also have an effect?

We agree this is possible, but given the scale at which birds use habitats (particularly more upland ones where there is likely to be greater roughness) we expect the effects to be small and they should be covered and correlated to habitat type.

L436: There appear to be two reference lists.

This is because the journal format requires references in the methods to be numbered sequentially after those in the main text, so we have simply kept them separate for now for ease of formatting later.

Reviewer #2 (Remarks to the Author):

This is an exceptionally well-written and clear study exploring the association between protected area (PA) designation and the state, change, community structure, and demography of birds in the UK. The topic is of relevance in light of the currently ongoing negotiations for a post-2020 biodiversity framework, where some are pushing for expanding the PA coverage target to 30 % by 2030. Clearly, we urgently need more evidence of the role of PAs and how effective PAs are as a policy tool to protect habitats and avoid population declines and extinctions. Most such studies have used forest loss or land cover change as a proxy for PA effectiveness, evaluating the impact PAs have in avoiding land use change. More studies are needed that look at the effect on species, communities, and population trends. The submitted study therefore sounded extremely interesting based on the abstract.

We thank the referee for recognising the “exceptionally well-written and clear” nature of the study and its topical relevance.

10The authors use Atlas based data on birds in the UK to develop a wide set of indicators for population viability: exploring both state, change, community structure, and even demography. This is interesting.

We appreciate the referee's interest, and would note that in addition to the comprehensive Atlas dataset on occurrence we were able to develop a much richer analysis looking at abundance, and changes therein (by making use of the BBS dataset), which has been done more rarely. Further, and uniquely as far as we are aware, we are able to explore a potential mechanism for the PA effects using a standardised national ring-recapture dataset (CES) quantifying productivity.

However, unfortunately the study design in terms of evaluating PA effectiveness is not sufficient and is merely exploring associations, rather than attempting to infer cause-effect. Powerful methods to create quasi-experimental setups are available and applying matching to the dataset could result in some interesting results. The need to account for the non-random location of PAs become apparent when looking at Figure S2, showing that PAs in the UK (as in most other countries in the world) are situated "high and far" (see seminal paper by Joppa & Pfaff 2009 calling for the need to account for this in PA effectiveness studies). There have been recent advances applying matching to population data (Terraube et al. 2020 (cited); Jellesmark et al. 2022 (preprint)), so should be doable with the data in the study as the atlas data is available from both inside and outside PAs. The likely most important confounding factors to account for would be accessibility and remoteness from urban centers (in addition to all commonly explored factors). Unfortunately, I think we have moved beyond the point of associative studies for inferring whether a policy intervention works or not, we need more robust study designs that can explicitly answer how well conservation interventions work (Ferraro & Hanauer. 2014; Baylis et al. 2016; Schleicher 2020).

We appreciate the referee's positive assessment of our work and thank them for the suggestion of extending it through statistical matching. While we agree that it can provide a way of removing some of the non-random variation in PA location and therefore can help overcome bias in sampling that can exist with studies such as this, we feel it does have some limitations compared to our initial regression approach, hence why we initially chose not to use it. Firstly, it requires a binary response, i.e. samples are either PA or not PA, whereas - because we are co-opting national monitoring scheme data - we have a continuum of sites from those entirely within a PA to those entirely outside, hence the regression approach we chose to better encapsulate the influence of partial PA coverage. Discretising the sites into a binary classification has the potential to lose significant power, and also asks a slightly different statistical question. Secondly, while the referee is right to note that while we, in general, have sufficient in and out of sample squares for the Atlas dataset, this becomes more problematic for the BBS and CES dataset where matching reduces our sample of species substantially. Importantly, this means that we are unable to use statistical matching approaches to generate

11counterfactuals for the rarest species, i.e. those for which PAs are most likely to be important; this is a significant limitation. It is, perhaps, also worth noting that statistical matching is only as good as the variables used to undertake the matching. Schleicher et al. (2020) note that matching is not a panacea as it assumes that there are no unobserved confounding variables and is less successful when there is substantial overlap between treated areas and confounding variables (i.e. a large region of common support), as is indeed the case here, and for which they recognise the necessity of alternative approaches. While statistical matching, therefore, has great potential in many cases, its benefits seem fewer in relation to the data we use. Given these issues, we retain our original analysis but do now include an additional matched analysis along the lines the referee suggests (L397-411); its conclusions largely support our original results (Fig S4, S5). By using both regression and statistical matching approaches, we hope our manuscript is much strengthened, allowing the reader to have much greater confidence in our results which are robust to variation in the approach adopted to answer the question.

We do fully agree, though, with the underlying point the referee makes about the need for more robust study designs to properly assess the effectiveness of conservation interventions. The CSM framework mentioned by Referee 1 provides one way of doing this within PA, although we sympathise with the difficulties experienced in implementing it. Here we are more interested in testing whether the PA network has an effect that is discernible through an (independent) national monitoring that is regularly used to inform national policy initiatives rather than trying to disentangle the effects of particular management on individual sites per se. But, ultimately we hope these results can be a starting point for a dialogue around that, with the analytical challenges we have alluded to providing insight into how such improvements might be made. This conversation might be especially topical given today's news highlighting moves to reconsider these designations <https://www.theguardian.com/environment/2022/jun/30/uk-government-scrap-european-law-protecting-special-habitats>

Further, we agree that distance from population center is an important confounding variable (though one which will be correlated with elevation and latitude, both of which we already included), so we have added a measure of human population density in the vicinity of each focal site to our species model (L360) and updated our results accordingly. It did not change our results substantively.

Minor comments:

line 8-9: The sentence "but the underpinning evidence for their effectiveness is mixed with causal connections rarely evaluated" gives the reader the impression the study will apply a counterfactual study design, i.e. evaluating what would have happened, had the PAs not been established.

12We disagree. This sentence is a statement of fact, we explicitly state in the next sentence that we provide an assessment of whether PA are associated with improved status (and we do provide some assessment of causal connections by, uniquely, looking at demographic effects), so we believe there is limited opportunity for confusion.

line 33-34: The phrasing of the third objective remains unclear for me: "iii) can they collectively facilitate the restoration or expansion of wider populations/habitats of conservation concern?"

We have reworded this sentence (L40) to clarify our meaning

line 52: please specify early on which national avifauna? Now I think one needs to go all the way to the Methods to see it written out that it is UK.

Done (L60)

line 55: perhaps include the current first draft of the new targets as a reference.
<https://www.cbd.int/doc/c/abb5/591f/2e46096d3f0330b08ce87a45/wg2020-03-03-en.pdf>

Reference added (L31, 64)

line 91: as said before the paper is exceptionally clear and well written, but here as a first time reader, I suggest the authors open up what they mean with "in areas with ", as that otherwise is only further explained in the methods and it would be good if the main text could stand on its own.

We have added some text at L98 to clarify this point at an early stage

Decision Letter, first revision:

5th September 2022

Dear Dr. Robinson,

Thank you for submitting your revised manuscript "Do conservation designations provide positive benefits for bird species and communities?" (NATECOLEVOL-220215790A). It has now been seen again by the original Reviewer 2, whose comments are below. We had been waiting to hear back from Reviewer 1 but have not received a review yet, and we felt able to proceed without that input. The reviewer find that the paper has improved in revision, and therefore we'll be happy in principle to

13publish it in Nature Ecology & Evolution, pending minor revisions to satisfy the reviewers' final requests and to comply with our editorial and formatting guidelines.

Please note that if we feel Reviewer 2's remaining concerns have been adequately addressed, particularly regarding providing further detail on the matching analysis, we hope not to need to go back to that reviewer again. However, if we are not sure about this, we may seek further review from that reviewer.

[REDACTED]

Reviewer #2 (Remarks to the Author):

I thank the authors for all the extra work they have done to improve the manuscript based on the previous round of review. They have added a matching approach to analyze the data based on my previous critique that the modelling did not account for the non-random location of PAs. I like the idea of using the two analyses to validate the results and I think this is a really interesting paper utilizing a unique dataset and for a country where the discussion about PA effectiveness is sorely needed.

I have only a few rather minor comments on the new version.

line 35 and throughout the text: Usually the abbreviation for Protected Area is PA, and for Protected Areas PAs. Now the text uses PA for the plural, but in one instance PAs (line 397). I suggest using PAs for the plural.

line 59: why the change in reference here? I think the original review from 2013 is better to support the claim, whereas the 2019 paper actually explores the link between PA and conservation outcomes.

line 63-64: This part now feels repetitive: "and there have been calls for an increased target of 30% coverage by 2030" as the 30 by 30 % goal was added already earlier (line 31-35). Perhaps just delete here and move references up, or apply some slight rephrasing to make the order of the text flow better.

line 397-411: for matching, the explanation of how you applied the method is now very short. Please add further detail on whether you matched with or without replacement, and the percentage of

14treated units that you found good matches for, and if you used a caliper.

line 428-431: Please give the number of species omitted for these reasons.

Fig. S3: There are some covariates for which there is only one observation, is that because the hollow and black dots are overlapping, so only the hollow one is visible, or is there some problem with the figures? (log_popdens and FW in A and B panels).

Fig. S3. Only here does it say that "Values below 0.25 (vertical line) were considered well matched", does that mean 0.25 was used as a caliper and values above this was dropped from analyses and if so, what percentage was this of all treated units? This is commonly reported for matching studies => add this information to the methods.

Comparing Fig. 1 and Fir. S4, what surprises me is the sample size...it seems it is higher for the matching than for you original analyses for the Range characteristics (3 first groups), why is that? => see comment above if some treatment units had to be dropped because you did not find good enough matches.

Finally, a paper worth citing and perhaps useful for the matching analyses, is the paper just out by Hannah Wauchope in Nature on "Protected areas have a mixed impact on waterbirds, but management helps": <https://www.nature.com/articles/s41586-022-04617-0>

Our ref: NATECOLEVOL-220215790A

13th September 2022

Dear Dr. Robinson,

Thank you for your patience as we've prepared the guidelines for final submission of your Nature Ecology & Evolution manuscript, "Do conservation designations provide positive benefits for bird species and communities?" (NATECOLEVOL-220215790A). Please carefully follow the step-by-step instructions provided in the attached file, and add a response in each row of the table to indicate the changes that you have made. Please also check and comment on any additional marked-up edits we have proposed within the text. Ensuring that each point is addressed will help to ensure that your revised manuscript can be swiftly handed over to our production team.

**We would like to start working on your revised paper, with all of the requested files and forms, as soon as possible (preferably within two weeks). Please get in contact with us immediately if you

15anticipate it taking more than two weeks to submit these revised files.**

In recognition of the time and expertise our reviewers provide to Nature Ecology & Evolution's editorial process, we would like to formally acknowledge their contribution to the external peer review of your manuscript entitled "Do conservation designations provide positive benefits for bird species and communities?". For those reviewers who give their assent, we will be publishing their names alongside the published article.

Nature Ecology & Evolution offers a Transparent Peer Review option for new original research manuscripts submitted after December 1st, 2019. As part of this initiative, we encourage our authors to support increased transparency into the peer review process by agreeing to have the reviewer comments, author rebuttal letters, and editorial decision letters published as a Supplementary item. When you submit your final files please clearly state in your cover letter whether or not you would like to participate in this initiative. Please note that failure to state your preference will result in delays in accepting your manuscript for publication.

Cover suggestions

As you prepare your final files we encourage you to consider whether you have any images or illustrations that may be appropriate for use on the cover of Nature Ecology & Evolution.

Nature Ecology & Evolution has now transitioned to a unified Rights Collection system which will allow our Author Services team to quickly and easily collect the rights and permissions required to publish

16your work. Approximately 10 days after your paper is formally accepted, you will receive an email in providing you with a link to complete the grant of rights. If your paper is eligible for Open Access, our Author Services team will also be in touch regarding any additional information that may be required to arrange payment for your article.

Please note that *Nature Ecology & Evolution* is a Transformative Journal (TJ). Authors may publish their research with us through the traditional subscription access route or make their paper immediately open access through payment of an article-processing charge (APC). Authors will not be required to make a final decision about access to their article until it has been accepted. [Find out more about Transformative Journals](https://www.springernature.com/gp/open-research/transformative-journals)

Authors may need to take specific actions to achieve [compliance with funder and institutional open access mandates](https://www.springernature.com/gp/open-research/funding/policy-compliance-faqs). If your research is supported by a funder that requires immediate open access (e.g. according to [Plan S principles](https://www.springernature.com/gp/open-research/plan-s-compliance)) then you should select the gold OA route, and we will direct you to the compliant route where possible. For authors selecting the subscription publication route, the journal's standard licensing terms will need to be accepted, including [self-archiving-and-license-to-publish](https://www.nature.com/nature-portfolio/editorial-policies/self-archiving-and-license-to-publish). Those licensing terms will supersede any other terms that the author or any third party may assert apply to any version of the manuscript.

Please use the following link for uploading these materials:
[REDACTED]

[REDACTED]

Reviewer #1:
None

Reviewer #2:

Remarks to the Author:

I thank the authors for all the extra work they have done to improve the manuscript based on the previous round of review. They have added a matching approach to analyze the data based on my previous critique that the modelling did not account for the non-random location of PAs. I like the idea of using the two analyses to validate the results and I think this is a really interesting paper utilizing a unique dataset and for a country where the discussion about PA effectiveness is sorely needed.

I have only a few rather minor comments on the new version.

line 35 and throughout the text: Usually the abbreviation for Protected Area is PA, and for Protected Areas PAs. Now the text uses PA for the plural, but in one instance PAs (line 397). I suggest using PAs for the plural.

line 59: why the change in reference here? I think the original review from 2013 is better to support the claim, whereas the 2019 paper actually explores the link between PA and conservation outcomes.

line 63-64: This part now feels repetitive: "and there have been calls for an increased target of 30% coverage by 2030" as the 30 by 30 % goal was added already earlier (line 31-35). Perhaps just delete here and move references up, or apply some slight rephrasing to make the order of the text flow better.

line 397-411: for matching, the explanation of how you applied the method is now very short. Please add further detail on whether you matched with or without replacement, and the percentage of treated units that you found good matches for, and if you used a caliper.

line 428-431: Please give the number of species omitted for these reasons.

Fig. S3: There are some covariates for which there is only one observation, is that because the hollow and black dots are overlapping, so only the hollow one is visible, or is there some problem with the figures? (log_popdens and FW in A and B panels).

Fig. S3. Only here does it say that "Values below 0.25 (vertical line) were considered well matched", does that mean 0.25 was used as a caliper and values above this was dropped from analyses and if so, what percentage was this of all treated units? This is commonly reported for matching studies => add this information to the methods.

Comparing Fig. 1 and Fir. S4, what surprises me is the sample size...it seems it is higher for the matching than for you original analyses for the Range characteristics (3 first groups), why is that? => see comment above if some treatment units had to be dropped because you did not find good enough matches.

Finally, a paper worth citing and perhaps useful for the matching analyses, is the paper just out by Hannah Wauchope in Nature on "Protected areas have a mixed impact on waterbirds, but management helps": <https://www.nature.com/articles/s41586-022-04617-0>

18Author Rebuttal, first revision:

We hereby provide a point by point response to the reviewer's comments. All line references refer to the track-changed version of the document.

Reviewer #2:

Remarks to the Author:

I thank the authors for all the extra work they have done to improve the manuscript based on the previous round of review. They have added a matching approach to analyze the data based on my previous critique that the modelling did not account for the non-random location of PAs. I like the idea of using the two analyses to validate the results and I think this is a really interesting paper utilizing a unique dataset and for a country where the discussion about PA effectiveness is sorely needed.

I have only a few rather minor comments on the new version.

line 35 and throughout the text: Usually the abbreviation for Protected Area is PA, and for Protected Areas PAs. Now the text uses PA for the plural, but in one instance PAs (line 397). I suggest using PAs for the plural.

Done, throughout.

line 59: why the change in reference here? I think the original review from 2013 is better to support the claim, whereas the 2019 paper actually explores the link between PA and conservation outcomes.

Done, line 62

line 63-64: This part now feels repetitive: "and there have been calls for an increased target of 30% coverage by 2030" as the 30 by 30 % goal was added already earlier (line 31-35). Perhaps just delete here and move references up, or apply some slight rephrasing to make the order of the text flow better.

Done, now reads "Designating PAs is a relatively straightforward policy tool to address biodiversity losses; implementing these effectively is, of course, a different matter (Stokstad et al. 2021)." (lines 66-69)

line 397-411: for matching, the explanation of how you applied the method is now very short. Please add further detail on whether you matched with or without replacement, and the percentage of treated units that you found good matches for, and if you used a caliper.

We have added more details in the methods text to clarify that we used matching without replacement and without calipers and have indicated the % squares matched (lines 408-417).

line 428-431: Please give the number of species omitted for these reasons.

More detail has been provided about numbers of species excluded for each reason. With very few exceptions, species with over-dispersion were also ones with extreme coefficients (lines 436-450).

Fig. S3: There are some covariates for which there is only one observation, is that because the hollow and black dots are overlapping, so only the hollow one is visible, or is there some problem with the figures? (log_popdens and FW in A and B panels).

Added sentence to legend: “In some cases the balance of samples was the same before and after matching so only one dot is visible.”

Fig. S3. Only here does it say that “Values below 0.25 (vertical line) were considered well matched”, does that mean 0.25 was used as a caliper and values above this was dropped from analyses and if so, what percentage was this of all treated units? This is commonly reported for matching studies => add this information to the methods.

The value of 0.25 is an indicator of how well the matching accounts for sample imbalance, which is good for many variables but not all. We did not drop any variables but rather included covariate weighting to account for any failure of matching and have added text in the methods to this effect (lines 416-417).

Comparing Fig. 1 and Fir. S4, what surprises me is the sample size...it seems it is higher for the matching than for you original analyses for the Range characteristics (3 first groups), why is that? => see comment above if some treatment units had to be dropped because you did not find good enough matches.

Thank you for pointing out this inconsistency. This was the result of inadvertently using different estimation algorithms for some matched and unmatched models, which resulted in differences in which species models converged successfully. We have now ensured that matched models for colonisation, occupation and persistence use the same fitting algorithm as the original models. As a result the sample sizes have changed slightly, but the overall conclusions remain very similar.

Finally, a paper worth citing and perhaps useful for the matching analyses, is the paper just out by Hannah Wauchope in Nature on "Protected areas have a mixed impact on waterbirds, but management helps": <https://www.nature.com/articles/s41586-022-04617-0>

We've already referenced this valuable paper on line 252 but have added it to the methods (line 405)?

Final Decision Letter:

11th October 2022

Dear Dr Robinson,

We are pleased to inform you that your Article entitled "Rare and declining bird species benefit most from designating protected areas for conservation in the UK", has now been accepted for publication in Nature Ecology & Evolution.

Over the next few weeks, your paper will be copyedited to ensure that it conforms to Nature Ecology and Evolution style. Once your paper is typeset, you will receive an email with a link to choose the appropriate publishing options for your paper and our Author Services team will be in touch regarding any additional information that may be required

You will not receive your proofs until the publishing agreement has been received through our system

Due to the importance of these deadlines, we ask you please us know now whether you will be difficult to contact over the next month. If this is the case, we ask you provide us with the contact information (email, phone and fax) of someone who will be able to check the proofs on your behalf, and who will be available to address any last-minute problems . Once your paper has been scheduled for online publication, the Nature press office will be in touch to confirm the details.

Acceptance of your manuscript is conditional on all authors' agreement with our publication policies (see www.nature.com/authors/policies/index.html). In particular your manuscript must not be published elsewhere and there must be no announcement of the work to any media outlet until the publication date (the day on which it is uploaded onto our web site).

Please note that *Nature Ecology & Evolution* is a Transformative Journal (TJ). Authors may publish their research with us through the traditional subscription access route or make their paper

21immediately open access through payment of an article-processing charge (APC). Authors will not be required to make a final decision about access to their article until it has been accepted. [Find out more about Transformative Journals](https://www.springernature.com/gp/open-research/transformative-journals)

Authors may need to take specific actions to achieve [compliance with funder and institutional open access mandates](https://www.springernature.com/gp/open-research/funding/policy-compliance-faqs). If your research is supported by a funder that requires immediate open access (e.g. according to [Plan S principles](https://www.springernature.com/gp/open-research/plan-s-compliance)) then you should select the gold OA route, and we will direct you to the compliant route where possible. For authors selecting the subscription publication route, the journal's standard licensing terms will need to be accepted, including [self-archiving and license to publish](https://www.nature.com/nature-portfolio/editorial-policies/self-archiving-and-license-to-publish). Those licensing terms will supersede any other terms that the author or any third party may assert apply to any version of the manuscript.

We welcome the submission of potential cover material (including a short caption of around 40 words) related to your manuscript; suggestions should be sent to Nature Ecology & Evolution as electronic files (the image should be 300 dpi at 210 x 297 mm in either TIFF or JPEG format). Please note that such pictures should be selected more for their aesthetic appeal than for their scientific content, and that colour images work better than black and white or grayscale images. Please do not try to design a cover with the Nature Ecology & Evolution logo etc., and please do not submit composites of images related to your work. I am sure you will understand that we cannot make any promise as to whether any of your suggestions might be selected for the cover of the journal.

To assist our authors in disseminating their research to the broader community, our SharedIt initiative

22provides you with a unique shareable link that will allow anyone (with or without a subscription) to read the published article. Recipients of the link with a subscription will also be able to download and print the PDF.

You can generate the link yourself when you receive your article DOI by entering it here: <http://authors.springernature.com/share>.

[REDACTED]

P.S. Click on the following link if you would like to recommend Nature Ecology & Evolution to your librarian <http://www.nature.com/subscriptions/recommend.html#forms>

** Visit the Springer Nature Editorial and Publishing website at http://editorial-jobs.springernature.com?utm_source=ejp_NEcoE_email&utm_medium=ejp_NEcoE_email&utm_campaign=ejp_NEcoE for more information about our career opportunities. If you have any questions please click [here](mailto:editorial.publishing.jobs@springernature.com). **